# A dynamic generalized fuzzy multi-criteria croup decision making approach for green supplier segmentation

Do Anh Duc[1,2], Luu Huu Van[1], Vincent F. Yu[3], Shuo-Yan Chou[3], Ngo Van Hien[4], Ngo The Chi[5], Dinh Van Toan[1], Luu Quoc Dat [1] *

1 VNU University of Economics and Business, Vietnam National University, Hanoi, Vietnam, 2 National Economics University, Hanoi, Vietnam, 3 Department of Industrial Management, National Taiwan University of Science and Technology, Taipei, Taiwan, 4 Phenikaa University, Hanoi, Vietnam, 5 Academy of Finance, Hanoi, Vietnam

* Datluuquoc@gmail.com

**Data Availability Statement:** All relevant data are within the paper and its Supporting Information files.

## Abstract

Supplier selection and segmentation are crucial tasks of companies in order to reduce costs and increase the competitiveness of their goods. To handle uncertainty and dynamicity in the supplier segmentation problem, this research thus proposes a new dynamic generalized fuzzy multi-criteria group decision making (MCGDM) approach from the aspects of capability and willingness and with respect to environmental issues. The proposed approach defines the aggregated ratings of alternatives, the aggregated weights of criteria, and the weighted ratings by using generalized fuzzy numbers with the effect of time weight. Next, we determine the ranking order of alternatives via a popular centroid-index ranking approach. Finally, two case studies demonstrate the efficiency of the proposed dynamic approach. The applications show that the proposed appoach is effective in solving the MCGDM in vague environment.

## Introduction

Supplier segmentation is a step that follows supplier selection and plays an important role in organizations for reducing production costs and optimally utilizing resources. Enterprises classify their suppliers from a selected set into distinct groups with different needs, characteristics, and requirements in order to adopt an appropriate strategic approach for handling different supplier segments [1]. Supplier segmentation is a highly complex decision-making problem that must consider many potential criteria and decision makers under a vague environment [2, 3]. Consequently, supplier segmentation can be viewed as a fuzzy multi-criteria group decision making (MCGDM) problem.

Numerous studies in the literature have proposed fuzzy multi-criteria decision making (MCDM) approaches to select and evaluate (green/sustainable) suppliers, with some recent applications found in [4–10]. While several studies used multi-criteria methods and fuzzy logic systems for solving supplier segmentation problem [2, 3, 11–13], existing studies on segmenting suppliers have paid limited attention to environmentally and socially related criteria [11]. Additionally, few studies have applied generalized fuzzy numbers (GFNs) to select or

**Funding:** This research is funded by "VNU University of Economics and Business, Vietnam National University, Hanoi" and "Korea Foundation for Advanced Studies (KFAS) and the Asia Research Center, Vietnam National University, Hanoi (ARC-VNU)" under project number CA.18.2A. This research was completed during and after the stay of the seventh author at the Vietnam Institute for Advanced Study in Mathematics (VIASM).

**Competing interests:** The authors have declared that no competing interests exist.

segment suppliers. Furthermore, they all have converted GFNs into normal fuzzy numbers through a normalization process and then applied fuzzy MCDM methods for normal fuzzy numbers. Nevertheless, the normalization process has a serious disadvantage—that is, the loss of information [14].

Chen [15] indicated in many practical situations that it is not possible to restrict the membership function to the normal form. Furthermore, the existing studies targeting supplier selection and segmentation only address static evaluation information for a certain period. However, in many real-life problems the decision makers are generally provided the information over different periods [16, 17]. Lee et al. [16] proposed a dynamic fuzzy MCGDM method for performance evaluation, while Mehdi et al. [17] presented a new fuzzy dynamic MCGDM approach to assess a subcontractor. Overall, it seems that no study has yet to propose a dynamic MCGDM using GFNs for solving the green supplier segmentation (GSS) problem with the effect of a time weight.

This study primarily proposes a new dynamic generalized fuzzy MCGDM approach from the aspects of capability and willingness with respect to environmental issues. The proposed approach defines the aggregated ratings of alternatives, the aggregated weights of criteria, and the aggregated weighted ratings using GFNs with the effect of time weight. We then determine the ranking order of alternatives via a popular centroid-index ranking approach proposed by [18]. Finally, two case studies demonstrate the efficiency of the proposed approach.

## Literature review on methods and criteria for supplier segmentation

This section presents an overview of the methods and criteria that have been used for supplier segmentation in the existing literature.

### Supplier segmentation methods

Supplier segmentation models have been widely explored ever since the pioneering works of [19, 20], who specified the variables required for segmenting suppliers [2, 3, 21–26]. Some of these models have been reviewed and discussed in the works of [20, 27–29]. Kraljic [20] presented a comprehensive portfolio approach to purchasing and supply segmentation. To classify materials or components, Kraljic [20] utilized two variables, the profit impact of a given item and the supply risk, under high and low levels that yield four segments: (1) non-critical items (supply risk: low; profit impact: low), (2) leverage items, (supply risk: low; profit impact: high), (3) bottleneck items (supply risk: high; profit impact: low), and (4) strategic items (supply risk: high; profit impact: high). Dyer et al. [30] developed strategic supplier segmentation based on the differences between outsourcing strategies. According to them, firms should maintain high levels of communication with suppliers that provide strategic inputs that contribute to the differential advantage of the buyer's final product. On the other hand, firms do not need to allocate significant resources to manage and work with suppliers that provide non-strategic inputs. Kaufman et al. [26] developed a strategic supplier typology that explains the differences in the composition and performance of various types of suppliers, using technology and collaboration to segment suppliers.

Svensson [27] applied three principal components, including the source of disturbance, the category of disturbance, and the type of logistics flow, in supplier segmentation. Hallikas et al. [24] described supplier and buyer dependency risks as the variables for classifying supplier relationships. Day et al. [28] presented the taxonomy of segmentation bases in which the buyer assesses the supply base from a purchasing perspective. Che [22] proposed two optimization mathematical models for the clustering and selection of suppliers. Model 1 is based on

customer demands to cluster suppliers under a minimal total within cluster variation. Model 2 takes the results of Model 1 to determine the optimal supplier combination based on quantity discount and customer demands. Rezaei & Ortt [31] proposed a framework for classifying suppliers based on supplier capabilities and willingness. Using their framework, it is possible to segment suppliers using multiple criteria, but most existing methods are based on just two criteria.

Rezaei et al. [32] presented an approach for segmenting and developing suppliers using capabilities and willingness criteria. They employed the best worst method (BWM) to define the relative weight of the criteria and further applied a scatter plot to segment the suppliers, where the horizontal and vertical axes are capabilities and willingness, respectively. Segura & Maroto [21] utilized a hybrid MCDM approach based on PROMETHEE and Multi-Attribute Utility Theory and used Analytic Hierarchy Process (AHP) for eliciting the weights of the criteria. The authors further took historical and reliable indicators to classify suppliers. Bai et al. [11] presented a novel methodology based on the rough set theory, VIKOR, and fuzzy C-means for green supplier segmentation, employing the dimensions of willingness and capabilities in their approach. Aineth & Ravindran [8] proposed a quantitative framework for sustainable procurement using the criteria of economic, environmental, and social hazards. Rezaei & Lajimi [33] combined purchasing portfolio matrix, supplier potential matrix, and BWM to segment suppliers. Appendix A in S1 Appendix compares the existing methods for supplier segmentation.

Supplier segmentation is a MCGDM problem that includes many criteria and decision makers within a vague environment. However, only a few studies in the literature applied the multi-criteria method and fuzzy logic systems to segment suppliers. Additionally, previous studies were limited to using normal fuzzy numbers and addressing the static evaluation information at a certain period to segment suppliers. Rezaei & Ortt [2] utilized the fuzzy AHP approach to segment suppliers using their capabilities and willingness criteria. Haghighi & Salahi [13] used the integrated fuzzy AHP approach and c-means algorithm to cluster suppliers. Akman [34] proposed a hybrid approach, including VIKOR, confirmatory factor analysis, and fuzzy c-means, to evaluate and segment suppliers in an automobile manufacturing company. The criteria of suppliers' capability and willingness were used to cluster suppliers. Lo & Sudjatmika [12] presented a modified fuzzy AHP approach for evaluating suppliers using bell-shaped membership functions. To our knowledge, no prior studies have developed the dynamic generalized fuzzy MCGDM approach with respect to environmental issues for solving supplier segmentation problem.

**Green supplier segmentation criteria.** Identifying the GSS criteria is one of the main challenges of a business enterprise to formulate proper supplier segmentation. To conduct GSS, several economic, environmental, and social dimensions should be considered [6], yet the majority of prior research only considered the evaluation criteria from the economic aspect. To segment the suppliers, our study's proposed approach takes into account not only economic criteria, but also environmental and social criteria. Appendix A in S1 Appendix summarizes the capabilities and willingness criteria drawing the greatest attention in recent literature.

## Establishment of a new approach for solving green supplier selection and segmentation

This section develops a new generalized fuzzy dynamic MCGDM approach to solve the green supplier selection and segmentation problem. The procedure of the proposed approach is described as follows.

### Identifying the green capabilities and willingness criteria

A committee of $k$ decision makers ($D_v, v = 1, \ldots, k$) is assumed responsible for evaluating $m$ suppliers ($A_i, i = 1, \ldots, m$) under $n$ selection criteria ($C_j, j = 1, \ldots, n$) in time sequence $t_u, u = 1, \ldots, h$, where the ratings of green suppliers versus each criterion and the importance weight of the criteria are expressed by using GTFN. The criteria are classified into two categories: capabilities ($C_j, j = 1, \ldots, l$) and willingness ($C_j, j = l+1, \ldots, n$).

A dynamic MCGDM approach can be concisely expressed in matrix format as:

$$C_1(t_u)\ C_2(t_u) \cdots C_j(t_u)$$

$$D_v(t_u) = \begin{matrix} A_1(t_u) \\ A_2(t_u) \\ \vdots \\ A_i(t_u) \end{matrix} \begin{bmatrix} x_{11}(t_u)x_{12}(t_u) \cdots x_{1j}(t_u) \\ x_{21}(t_u)x_{22}(t_u) \cdots x_{2j}(t_u) \\ \vdots\ \vdots\ \vdots\ \vdots \\ x_{i1}(t_u)x_{i2}(t_u) \cdots x_{ij}(t_u) \end{bmatrix}$$

### Aggregating the importance weights of the criteria

Let $w_{jv}(t_u) = \langle o_{jv}(t_u), p_{jv}(t_u), q_{jv}(t_u); \varpi_{jv}(t_u) \rangle, w_{jv}(t_u) \in R^*, j = 1, \ldots, n, v = 1, \ldots, k, u = 1, \ldots, h$, be the weight assigned by decision maker $D_v$ to criterion $C_j(C_j, j = 1, \ldots, n)$ in time sequence $t_u$. The average weight, $w_j = (o_j, p_j, q_j; \varpi_j)$, of criterion $C_j$ assessed by the committee of $k$ decision makers can be evaluated as:

$$w_j = \frac{1}{h*k} \otimes \langle w_{j1}(t_1) \oplus w_{j2}(t_2) \oplus \ldots \oplus w_{jk}(t_u) \rangle, \tag{1}$$

where $o_j = \frac{1}{h*k}\sum_{v=1}^{k} o_{jv}(t_u), p_j = \frac{1}{h*k}\sum_{v=1}^{k} p_{jv}(t_u), q_j = \frac{1}{h*k}\sum_{v=1}^{k} q_{jv}(t_u)$ and $\varpi_j = \min\{\varpi_{j1}(t_1), \varpi_{j2}(t_2), \ldots, \varpi_{jk}(t_u)\}$.

### Aggregating the ratings of green suppliers versus the criteria

Let $x_{ijv}(t_u) = \langle e_{ijv}(t_u), f_{ijv}(t_u), g_{ijv}(t_u); \varpi_{ijv}(t_u) \rangle, i = 1, \ldots, m, j = 1, \ldots, n, v = 1, \ldots, k, u = 1, \ldots, h$, be the suitability ratings assigned to the green suppliers $A_i$, by decision makers $D_v$ for criteria $C_j$ in time sequence $t_u$. The averaged suitability ratings, $x_{ij} = (e_{ij}, f_{ij}, g_{ij}; \varpi_{ij})$, can be evaluated as:

$$x_{ij} = \frac{1}{h*k} \otimes (x_{ij1}(t_1) \oplus x_{ij2}(t_2) \oplus \ldots \oplus x_{ijv}(t_u) \oplus \ldots \oplus x_{ijk}(t_h)), \tag{2}$$

where $e_{ij} = \frac{1}{h*k}\sum_{v=1}^{k} e_{ijv}(t_u), f_{ij} = \frac{1}{h*k}\sum_{v=1}^{k} f_{ijv}(t_u), g_{ij} = \frac{1}{h*k}\sum_{v=1}^{k} g_{ijv}(t_u)$, and $\varpi_{ij} = \min(\varpi_{ij1}(t_1), \varpi_{ij2}(t_2), \ldots, \varpi_{ijk}(t_h)\}$.

### Constructing the weighted fuzzy decision matrix

The weighted decision matrices $S_{i1} = (d_{i1}, h_{i1}, i_{i1}; \varpi_{i1})$ and $S_{i2} = (d_{i2}, h_{i2}, i_{i2}; \varpi_{i2})$ of the green suppliers $A_i$ versus the capabilities ($C_j, j = 1, \ldots, l$) and willingness criteria ($C_j, j = l+1, \ldots, n$) in time $t_u$ are respectively defined as follows:

$$S_{i1} = \frac{1}{l}\sum_{j=1}^{l} (s_{ij})_{m.l} = \frac{1}{l}\sum_{j=1}^{l} x_{ij} \otimes w_j, i = 1, \ldots, m; j = 1, \ldots, l, \tag{3}$$

$$S_{i2} = \frac{1}{n-l-1}\sum_{j=l+1}^{n}(s_{ij})_{m.(n-l)} = \frac{1}{n-l-1}\sum_{j=l+1}^{n}x_{ij}\otimes w_j, \quad i=1,\ldots,m; j=l+1,\ldots,n. \quad (4)$$

## Defuzzification

This study applies the popular centroid-index ranking approach proposed by [18] to determine the distance values between the centroid and minimum points of green suppliers versus the capabilities and willingness criteria.

## Segmenting the green suppliers

Based on the distance values between the centroid and minimum points of the green suppliers in defuzzification process versus the capabilities and willingness criteria, we divide the green suppliers into $2 \times 2$ segments, including Group 1 (low capabilities and low willingness), Group 2 (low capabilities and high willingness), Group 3 (high capabilities and low willingness), and Group 4 (high capabilities and high willingness). The cut-off points, which are the potential values of the distance, are determined by the decision makers; i.e., all decision makers give the linguistic variables for the ratings of alternatives as Fair = (0.3, 0.5, 0.7; 0.8).

# Implementation of the proposed dynamic generalized fuzzy MCGDM approach

This section applies the proposed approach in the case of a medium-sized transport equipment company located in northern Vietnam. The managers of this company have become perplexed on how to effectively manage their suppliers to maximize their profit due to the increase in the number of suppliers. We apply the proposed approach to the process of this firm's green supplier segmentation to help it segment its suppliers and test the efficacy of the proposed method. Data were collected by conducting semi-structured interviews with the company's top managers and department heads (decision-makers). Three decision makers ($D_1$, $D_2$, and $D_3$) were requested to separately evaluate the importance weights of the capabilities and willingness criteria and the ratings of GSS at three different times ($t_1$, $t_2$, and $t_3$). We characterize the entire GSS procedure by the following steps.

*Step 1*: Aggregate the importance weights of the respective capabilities and willingness criteria.

*Step 2*: Aggregate the ratings of green suppliers versus capabilities and willingness criteria, respectively.

*Step 3*: Construct the weighted fuzzy decision matrices.

*Step 4*: Calculation of the distance of each green supplier.

*Step 5*: Segment the green suppliers.

Steps 1 and 2 were performed by the company's managers (i.e., the three decision-makers $D_1$, $D_2$, and $D_3$) without any intervention from the authors. Steps 3 to 5 were calculated using the proposed approach.

## Aggregation of the importance weights of the respective green capabilities and willingness criteria

Following the review of the literature and discussions with the top managers and department heads, we select six capabilities (i.e., price/cost—$C_1$, delivery—$C_2$, quality—$C_3$, reputation and

**Table 1. Aggregated weights of the criteria evaluated by the decision makers.**

| Criterion | Decision maker | | | | | | | | | $w_{ij}$ |
|---|---|---|---|---|---|---|---|---|---|---|
| | $t_1$ | | | $t_2$ | | | $t_3$ | | | |
| | $D_1$ | $D_2$ | $D_3$ | $D_1$ | $D_2$ | $D_3$ | $D_1$ | $D_2$ | $D_3$ | |
| $C_1$ | VI | VI | VI | AI | VI | AI | AI | VI | AI | (0.633, 0.789, 0.944; 0.900) |
| $C_2$ | VI | I | I | I | I | I | VI | VI | I | (0.433, 0.567, 0.700; 0.800) |
| $C_3$ | VI | AI | VI | AI | VI | VI | VI | VI | VI | (0.567, 0.744, 0.922; 0.900) |
| $C_4$ | VI | VI | AI | VI | VI | VI | I | VI | I | (0.511, 0.678, 0.844; 0.800) |
| $C_5$ | AI | VI | VI | I | VI | I | I | VI | I | (0.489, 0.633, 0.778; 0.800) |
| $C_6$ | I | VI | I | I | VI | VI | I | VI | VI | (0.456, 0.611, 0.767; 0.800) |
| $W_1$ | I | I | I | VI | I | I | I | VI | I | (0.422, 0.544, 0.667; 0.800) |
| $W_2$ | VI | I | VI | I | I | VI | VI | I | I | (0.444, 0.589, 0.733; 0.800) |
| $W_3$ | I | I | I | I | VI | I | I | VI | I | (0.422, 0.544, 0.667; 0.800) |
| $W_4$ | I | VI | I | I | VI | VI | VI | VI | I | (0.456, 0.611, 0.767; 0.800) |

position in industry—$C_4$, financial position—$C_5$, hazardous waste management—$C_6$) and four willingness criteria (i.e., commitment to quality—$W_1$, commitment to continuous improvement in product and process—$W_2$, relationship closeness—$W_3$, willingness to share information, ideas, technology, and cost savings—$W_4$) for evaluating and segmenting suppliers. After determining the green suppliers' criteria, the three company's managers are asked to define the level of importance of each criterion through a linguistic variable. Table 1 shows the aggregate weights of the criteria using Eq (1).

## Aggregation of the ratings of green suppliers versus the capabilities and willingness criteria

The decision makers define the suitability ratings of twelve green suppliers (i.e., $A_1, \ldots, A_{12}$) versus the capabilities and willingness criteria using the linguistic variables. Table 2A to 2E (in Appendix C in S1 Appendix) present the aggregated suitability ratings of the suppliers versus the six capabilities criteria (i.e., $C_1, \ldots, C_7$) and four willingness criteria (i.e., $W_1, \ldots, W_6$) from the three decision makers obtained from Eq (2) and Table 1 (in Appendix B in S1 Appendix).

**Table 2. Final fuzzy evaluation values of each supplier.**

| Supplier | Capabilities criteria | Willingness criteria |
|---|---|---|
| $A_1$ | (0,214, 0,405, 0,653; 0,700) | (0,126, 0,262, 0,443; 0,700) |
| $A_2$ | (0,124, 0,261, 0,444; 0,600) | (0,214, 0,387, 0,611; 0,800) |
| $A_3$ | (0,303, 0,507, 0,762; 0,800) | (0,198, 0,372, 0,598; 0,800) |
| $A_4$ | (0,131, 0,269, 0,453; 0,600) | (0,214, 0,391, 0,620; 0,800) |
| $A_5$ | (0,228, 0,422, 0,674; 0,700) | (0,191, 0,358, 0,576; 0,700) |
| $A_6$ | (0,231, 0,428, 0,685; 0,700) | (0,219, 0,391, 0,611; 0,800) |
| $A_7$ | (0,298, 0,484, 0,716; 0,700) | (0,212, 0,386, 0,612; 0,800) |
| $A_8$ | (0,137, 0,286, 0,487; 0,600) | (0,130, 0,266, 0,449; 0,600) |
| $A_9$ | (0,231, 0,428, 0,683; 0,700) | (0,205, 0,377, 0,601; 0,800) |
| $A_{10}$ | (0,258, 0,448, 0,692; 0,600) | (0,184, 0,353, 0,575; 0,700) |
| $A_{11}$ | (0,239, 0,440, 0,699; 0,800) | (0,203, 0,378, 0,605; 0,800) |
| $A_{12}$ | (0,131, 0,273, 0,464; 0,600) | (0,214, 0,378, 0,589; 0,600) |

**Table 3. Distance measurement.**

| Supplier | Capabilities criteria | | Willingness criteria | |
|:---:|:---:|:---:|:---:|:---:|
| | Centroid point $A_i\,(\bar{x}_A, \bar{y}_A)$ | Distance $D(A_i, Go)$ | Centroid point $A_i\,(\bar{x}_A, \bar{y}_A)$ | Distance $D(A_i, Go)$ |
| $A_1$ | (0,424, 0,233) | 0,314 | (0,277, 0,233) | 0,177 |
| $A_2$ | (0,276, 0,200) | 0,172 | (0,404, 0,267) | 0,298 |
| $A_3$ | (0,524, 0,267) | 0,414 | (0,389, 0,267) | 0,284 |
| $A_4$ | (0,284, 0,200) | 0,179 | (0,409, 0,267) | 0,302 |
| $A_5$ | (0,442, 0,233) | 0,331 | (0,375, 0,233) | 0,266 |
| $A_6$ | (0,448, 0,233) | 0,338 | (0,407, 0,267) | 0,300 |
| $A_7$ | (0,499, 0,233) | 0,387 | (0,404, 0,267) | 0,297 |
| $A_8$ | (0,303, 0,200) | 0,197 | (0,282, 0,200) | 0,175 |
| $A_9$ | (0,447, 0,233) | 0,337 | (0,394, 0,267) | 0,288 |
| $A_{10}$ | (0,466, 0,200) | 0,351 | (0,370, 0,233) | 0,261 |
| $A_{11}$ | (0,459, 0,267) | 0,352 | (0,396, 0,267) | 0,290 |
| $A_{12}$ | (0,289, 0,200) | 0,184 | (0,394, 0,200) | 0,279 |

## Determination of the weighted rating

Table 2 shows the final fuzzy evaluation values of each green supplier using Eqs (3) and (4).

## Calculation of the distance of each green supplier

We obtain the distance between the centroid point and the minimum point Go = (0,124, 0,600) of each green supplier as depicted in Table 3 by using the data in Table 2 and the ranking approach proposed by [18].

## Segmentation of the suppliers

Based on the distance scores for the capabilities and willingness of each green supplier, we assign 12 green suppliers to one of four segments (Fig 1) using Step 6 of the proposed methodology. In this step, the cut-off points of the green supplier's capabilities and willingness are 0.2084 and 0.1814, respectively. Fig 1 and Table 4 show that one green supplier is assigned to Group 1, three green suppliers to Group 2, one green supplier to Group 3, and seven green suppliers to Group 4. Thus, the company has seven good green suppliers, but five of them lack capabilities, willingness, or both.

The results indicate that the company can use different strategies to handle various segments and may try and develop those green suppliers that are less capable and less willing to cooperate (i.e., Group 1 green suppliers) or terminate its relationship with them in favor of good alternatives [2, 3]. Group 2 green suppliers are willing to cooperate, but are less competent to meet the buyer's requirements. The company should help these green suppliers improve their capabilities and performance or replace them with capable ones in the short term [35]. Group 3 green suppliers have high capabilities, but exhibit a low-level willingness to cooperate. The company should focus on improving its relationship with these green suppliers and determine various approaches on how to become attractive to them [36]. Group 4 green suppliers, which are the best green suppliers of the company, have great capabilities and a high level of willingness. The company should maintain a close long-term relationship with these green suppliers [31].

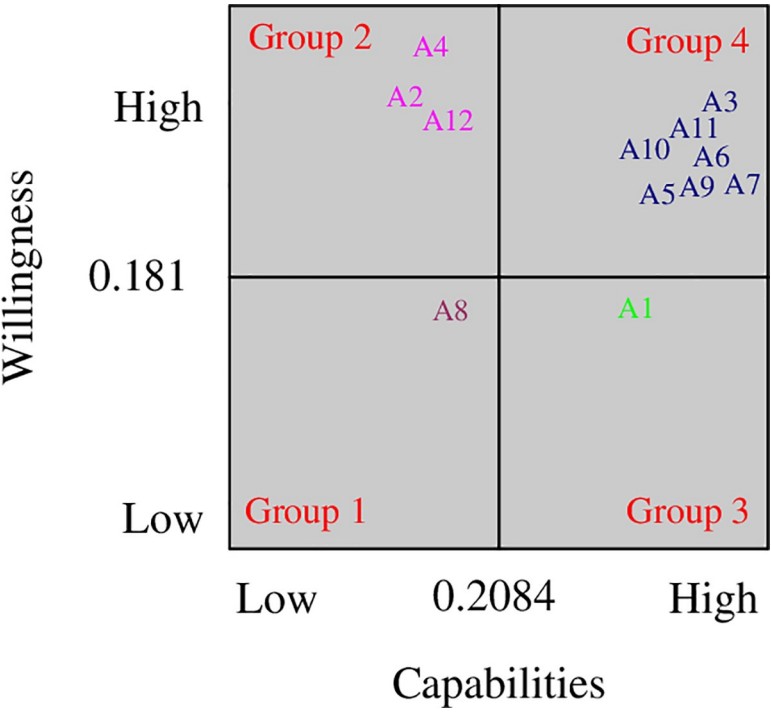

**Fig 1. Final supplier segmentation results.**

## Comparison of the proposed method with another fuzzy MCDM method

This section compares the proposed approach in time $t_u, u = 1$ with another fuzzy MCDM approach to demonstrate its advantages and applicability by reconsidering the example investigated by [2]. In this example, a medium-sized broiler (meat-type chicken) company in the food industry intends to segment its suppliers. Six criteria for capabilities and six criteria for willingness are selected to segment 43 suppliers based on the decision makers (i.e., the managers). Table 5 shows the importance weights of the capabilities and willingness criteria.

Table 6 demonstrates the averaged ratings of suppliers versus the capabilities and willingness criteria based on the data presented in Table 3 in the work of [2] and in Table 1 (in Appendix B in S1 Appendix) of this paper.

We obtain the distance between the centroid and minimum points of 43 suppliers by using the ranking approach proposed by [17] as denoted in Table 7.

Based on the distance scores for the capabilities and willingness of each supplier, we assign 43 suppliers to one of four segments using Step 7 of the proposed method. The cut-off points of the supplier's capabilities and willingness are 0.196 and 0.1996, respectively. Table 8 shows

**Table 4. Segments of the suppliers.**

| Segment | No. of suppliers | Supplier(s) |
|---|---|---|
| Group 1 | 1 | $A_8$ |
| Group 2 | 3 | $A_2, A_4, A_{12}$ |
| Group 3 | 1 | $A_1$ |
| Group 4 | 7 | $A_3, A_5, A_6, A_7, A_{09}, A_{10}, A_{11}$ |

**Table 5. Importance weights of the capabilities and willingness criteria.**

| Capabilities criterion | Fuzzy weight | Willingness criterion | Fuzzy weight |
|---|---|---|---|
| $C_1^C$ | (0.065, 0.106, 0.181; 1.0) | $C_1^W$ | (0.114, 0.206, 0.350; 1.0) |
| $C_2^C$ | (0.110, 0.161, 0.238; 1.0) | $C_2^W$ | (0.086, 0.150, 0.266; 1.0) |
| $C_3^C$ | (0.148, 0.206, 0.279; 1.0) | $C_3^W$ | (0.094, 0.150, 0.253; 1.0) |
| $C_4^C$ | (0.115, 0.161, 0.231; 1.0) | $C_4^W$ | (0.094, 0.150, 0.253; 1.0) |
| $C_5^C$ | (0.109, 0.161, 0.240; 1.0) | $C_5^W$ | (0.127, 0.206, 0.328; 1.0) |
| $C_6^C$ | (0.132, 0.206, 0.302; 1.0) | $C_6^W$ | (0.074, 0.137, 0.250; 1.0) |

that three suppliers are assigned to Group 1, nine suppliers to Group 2, three suppliers to Group 3, and twenty-eight suppliers to Group 4.

Table 8 shows a slight difference between the segments of the 43 suppliers using the proposed method and the approach introduced by [2, 3]. The reason for the difference is that the techniques proposed by [2, 3] use the crisp values to measure the ratings of the suppliers. This proceeding is unreasonable, because the supplier evaluation criteria include both quantitative and qualitative criteria. The proposed method herein employs GFNs to represent the ratings of suppliers.

## Discussions and conclusions

Green supplier segmentation (GSS) is a critical marketing activity for companies having many suppliers. Rather than formulating individual strategies for each supplier, companies can now adopt an appropriate strategic approach for handling different supplier segments. To manage

**Table 6. Average ratings of suppliers versus the capabilities and willingness criteria.**

| Supplier no. | Capabilities criteria | Willingness criteria | Supplier no. | Capabilities criteria | Willingness criteria |
|---|---|---|---|---|---|
| 1 | (0.037, 0.085, 0.170; 0.8) | (0.050, 0.116, 0.250; 0.8) | 23 | (0.051, 0.105, 0.199; 0.8) | (0.054, 0.122, 0.259; 0.8) |
| 2 | (0.051, 0.105, 0.197; 0.8) | (0.061, 0.128, 0.261; 0.8) | 24 | (0.024, 0.055, 0.112; 0.8) | (0.043, 0.105, 0.235; 0.8) |
| 3 | (0.052, 0.106, 0.200; 0.8) | (0.046, 0.110, 0.240; 0.8) | 25 | (0.039, 0.090, 0.181; 0.8) | (0.040, 0.102, 0.230; 0.8) |
| 4 | (0.058, 0.111, 0.204; 0.8) | (0.061, 0.130, 0.266; 0.8) | 26 | (0.037, 0.088, 0.179; 0.8) | (0.056, 0.123, 0.257; 0.8) |
| 5 | (0.041, 0.092, 0.185; 0.8) | (0.049, 0.112, 0.240; 0.8) | 27 | (0.046, 0.101, 0.197; 0.8) | (0.042, 0.105, 0.236; 0.8) |
| 6 | (0.039, 0.089, 0.176; 0.8) | (0.049, 0.113, 0.243; 0.8) | 28 | (0.058, 0.115, 0.211; 0.8) | (0.040, 0.100, 0.227; 0.8) |
| 7 | (0.056, 0.110, 0.203; 0.8) | (0.047, 0.109, 0.235; 0.8) | 29 | (0.033, 0.082, 0.169; 0.8) | (0.040, 0.100, 0.226; 0.8) |
| 8 | (0.063, 0.121, 0.219; 0.8) | (0.014, 0.057, 0.153; 0.8) | 30 | (0.019, 0.053, 0.115; 0.8) | (0.044, 0.104, 0.226; 0.8) |
| 9 | (0.017, 0.050, 0.109; 0.8) | (0.014, 0.057, 0.153; 0.8) | 31 | (0.039, 0.090, 0.181; 0.8) | (0.045, 0.107, 0.233; 0.8) |
| 10 | (0.017, 0.050, 0.109; 0.8) | (0.014, 0.057, 0.153; 0.8) | 32 | (0.052, 0.101, 0.183; 0.8) | (0.051, 0.117, 0.251; 0.8) |
| 11 | (0.043, 0.096, 0.189; 0.8) | (0.065, 0.133, 0.269; 0.8) | 33 | (0.045, 0.100, 0.195; 0.8) | (0.055, 0.123, 0.261; 0.8) |
| 12 | (0.048, 0.100, 0.188; 0.8) | (0.064, 0.133, 0.269; 0.8) | 34 | (0.046, 0.098, 0.189; 0.8) | (0.013, 0.053, 0.142; 0.8) |
| 13 | (0.054, 0.110, 0.207; 0.8) | (0.057, 0.121, 0.249; 0.8) | 35 | (0.046, 0.097, 0.186; 0.8) | (0.054, 0.122, 0.259; 0.8) |
| 14 | (0.031, 0.075, 0.154; 0.8) | (0.038, 0.098, 0.224; 0.8) | 36 | (0.039, 0.090, 0.181; 0.8) | (0.044, 0.107, 0.238; 0.8) |
| 15 | (0.043, 0.096, 0.189; 0.8) | (0.037, 0.092, 0.206; 0.8) | 37 | (0.061, 0.117, 0.212; 0.8) | (0.053, 0.122, 0.259; 0.8) |
| 16 | (0.025, 0.060, 0.124; 0.8) | (0.037, 0.095, 0.218; 0.8) | 38 | (0.044, 0.094, 0.182; 0.8) | (0.039, 0.100, 0.226; 0.8) |
| 17 | (0.025, 0.059, 0.119; 0.8) | (0.060, 0.128, 0.265; 0.8) | 39 | (0.038, 0.089, 0.180; 0.8) | (0.020, 0.068, 0.173; 0.8) |
| 18 | (0.014, 0.045, 0.101; 0.8) | (0.050, 0.117, 0.251; 0.8) | 40 | (0.047, 0.099, 0.191; 0.8) | (0.051, 0.117, 0.251; 0.8) |
| 19 | (0.052, 0.106, 0.201; 0.8) | (0.015, 0.057, 0.149; 0.8) | 41 | (0.032, 0.078, 0.160; 0.8) | (0.040, 0.100, 0.227; 0.8) |
| 20 | (0.039, 0.088, 0.175; 0.8) | (0.033, 0.090, 0.210; 0.8) | 42 | (0.053, 0.108, 0.202; 0.8) | (0.049, 0.112, 0.240; 0.8) |
| 21 | (0.019, 0.059, 0.133; 0.8) | (0.013, 0.052, 0.139; 0.8) | 43 | (0.031, 0.071, 0.142; 0.8) | (0.059, 0.125, 0.257; 0.8) |
| 22 | (0.048, 0.101, 0.193; 0.8) | (0.052, 0.117, 0.249; 0.8) | | | |

**Table 7. Distance measurement.**

| Supplier | Capabilities criteria | | | Willingness criteria | | |
|---|---|---|---|---|---|---|
| | Centroid point | Minimum point | Distance | Centroid point | Minimum point | Distance |
| 1 | (0.097, 0.333) | (0.014, 0.333) | 0,196 | (0.139, 0.333) | (0.013, 0.333) | 0,218 |
| 2 | (0.118, 0.333) | (0.014, 0.333) | 0,206 | (0.150, 0.333) | (0.013, 0.333) | 0,224 |
| 3 | (0.119, 0.333) | (0.014, 0.333) | 0,207 | (0.132, 0.333) | (0.013, 0.333) | 0,214 |
| 4 | (0.124, 0.333) | (0.014, 0.333) | 0,209 | (0.153, 0.333) | (0.013, 0.333) | 0,226 |
| 5 | (0.106, 0.333) | (0.014, 0.333) | 0,200 | (0.133, 0.333) | (0.013, 0.333) | 0,215 |
| 6 | (0.101, 0.333) | (0.014, 0.333) | 0,198 | (0.135, 0.333) | (0.013, 0.333) | 0,216 |
| 7 | (0.123, 0.333) | (0.014, 0.333) | 0,209 | (0.131, 0.333) | (0.013, 0.333) | 0,213 |
| 8 | (0.134, 0.333) | (0.014, 0.333) | 0,215 | (0.074, 0.333) | (0.013, 0.333) | 0,188 |
| 9 | (0.059, 0.333) | (0.014, 0.333) | 0,183 | (0.075, 0.333) | (0.013, 0.333) | 0,188 |
| 10 | (0.059, 0.333) | (0.014, 0.333) | 0,183 | (0.075, 0.333) | (0.013, 0.333) | 0,188 |
| 11 | (0.109, 0.333) | (0.014, 0.333) | 0,202 | (0.156, 0.333) | (0.013, 0.333) | 0,228 |
| 12 | (0.112, 0.333) | (0.014, 0.333) | 0,203 | (0.155, 0.333) | (0.013, 0.333) | 0,228 |
| 13 | (0.124, 0.333) | (0.014, 0.333) | 0,209 | (0.142, 0.333) | (0.013, 0.333) | 0,220 |
| 14 | (0.087, 0.333) | (0.014, 0.333) | 0,192 | (0.120, 0.333) | (0.013, 0.333) | 0,207 |
| 15 | (0.109, 0.333) | (0.014, 0.333) | 0,202 | (0.112, 0.333) | (0.013, 0.333) | 0,203 |
| 16 | (0.070, 0.333) | (0.014, 0.333) | 0,186 | (0.117, 0.333) | (0.013, 0.333) | 0,206 |
| 17 | (0.068, 0.333) | (0.014, 0.333) | 0,186 | (0.151, 0.333) | (0.013, 0.333) | 0,225 |
| 18 | (0.053, 0.333) | (0.014, 0.333) | 0,182 | (0.139, 0.333) | (0.013, 0.333) | 0,218 |
| 19 | (0.120, 0.333) | (0.014, 0.333) | 0,207 | (0.074, 0.333) | (0.013, 0.333) | 0,188 |
| 20 | (0.101, 0.333) | (0.014, 0.333) | 0,198 | (0.111, 0.333) | (0.013, 0.333) | 0,203 |
| 21 | (0.070, 0.333) | (0.014, 0.333) | 0,186 | (0.068, 0.333) | (0.013, 0.333) | 0,186 |
| 22 | (0.114, 0.333) | (0.014, 0.333) | 0,204 | (0.139, 0.333) | (0.013, 0.333) | 0,218 |
| 23 | (0.118, 0.333) | (0.014, 0.333) | 0,206 | (0.145, 0.333) | (0.013, 0.333) | 0,221 |
| 24 | (0.064, 0.333) | (0.014, 0.333) | 0,185 | (0.128, 0.333) | (0.013, 0.333) | 0,212 |
| 25 | (0.103, 0.333) | (0.014, 0.333) | 0,199 | (0.124, 0.333) | (0.013, 0.333) | 0,210 |
| 26 | (0.102, 0.333) | (0.014, 0.333) | 0,198 | (0.145, 0.333) | (0.013, 0.333) | 0,222 |
| 27 | (0.115, 0.333) | (0.014, 0.333) | 0,204 | (0.128, 0.333) | (0.013, 0.333) | 0,212 |
| 28 | (0.128, 0.333) | (0.014, 0.333) | 0,211 | (0.122, 0.333) | (0.013, 0.333) | 0,209 |
| 29 | (0.095, 0.333) | (0.014, 0.333) | 0,195 | (0.122, 0.333) | (0.013, 0.333) | 0,209 |
| 30 | (0.062, 0.333) | (0.014, 0.333) | 0,184 | (0.125, 0.333) | (0.013, 0.333) | 0,210 |
| 31 | (0.103, 0.333) | (0.014, 0.333) | 0,199 | (0.128, 0.333) | (0.013, 0.333) | 0,212 |
| 32 | (0.112, 0.333) | (0.014, 0.333) | 0,203 | (0.140, 0.333) | (0.013, 0.333) | 0,218 |
| 33 | (0.113, 0.333) | (0.014, 0.333) | 0,204 | (0.146, 0.333) | (0.013, 0.333) | 0,222 |
| 34 | (0.111, 0.333) | (0.014, 0.333) | 0,202 | (0.069, 0.333) | (0.013, 0.333) | 0,186 |
| 35 | (0.110, 0.333) | (0.014, 0.333) | 0,202 | (0.145, 0.333) | (0.013, 0.333) | 0,221 |
| 36 | (0.103, 0.333) | (0.014, 0.333) | 0,199 | (0.130, 0.333) | (0.013, 0.333) | 0,213 |
| 37 | (0.130, 0.333) | (0.014, 0.333) | 0,212 | (0.145, 0.333) | (0.013, 0.333) | 0,221 |
| 38 | (0.107, 0.333) | (0.014, 0.333) | 0,201 | (0.122, 0.333) | (0.013, 0.333) | 0,208 |
| 39 | (0.102, 0.333) | (0.014, 0.333) | 0,198 | (0.087, 0.333) | (0.013, 0.333) | 0,193 |
| 40 | (0.112, 0.333) | (0.014, 0.333) | 0,203 | (0.140, 0.333) | (0.013, 0.333) | 0,218 |
| 41 | (0.090, 0.333) | (0.014, 0.333) | 0,193 | (0.122, 0.333) | (0.013, 0.333) | 0,209 |
| 42 | (0.121, 0.333) | (0.014, 0.333) | 0,207 | (0.133, 0.333) | (0.013, 0.333) | 0,215 |
| 43 | (0.081, 0.333) | (0.014, 0.333) | 0,190 | (0.147, 0.333) | (0.013, 0.333) | 0,222 |

the uncertainty and dynamics of GSS, this study develops a new dynamic generalized fuzzy MCGDM using capabilities and willingness criteria. The proposed approach contributes to the body of GSS literature in four significant directions. First, it expands previous studies by using

**Table 8. Segments of the 43 suppliers.**

| Segment | No. of suppliers | Suppliers |
|---|---|---|
| Group 1 | 3 | $A_9$, $A_{10}$, and $A_{21}$ |
| Group 2 | 9 | $A_{14}$, $A_{16}$, $A_{17}$, $A_{18}$, $A_{24}$, $A_{29}$, $A_{30}$, $A_{41}$, and$A_{43}$ |
| Group 3 | 3 | $A_{19}$, $A_{34}$, and $A_{39}$ |
| Group 4 | 28 | $A_1$, $A_2$, $A_3$, $A_4$, $A_5$, $A_6$, $A_7$, $A_8$, $A_{11}$, $A_{12}$, $A_{13}$, and $A_{15}$, $A_{20}$, $A_{21}$, $A_{22}$, $A_{23}$, $A_{25}$, $A_{26}$, $A_{27}$, $A_{28}$, $A_{31}$, $A_{32}$, $A_{33}$, $A_{35}$, $A_{36}$, $A_{37}$, $A_{38}$, and $A_{40}$ |

GFNs instead of fuzzy numbers. Second, it is able to solve the supplier segmentation problem at different periods instead of one period. Third, it considers not only economic criteria, but also environmental and social criteria from the aspects of suppliers' capability and willingness. Fourth, the approach can solve the GSS problem and also be employed in other management problems under similar settings.

The proposed framework uses GFNs to express the aggregated ratings of alternatives, the aggregated importance weights of criteria, and the aggregated weighted ratings with the effect of time weight. In order to rank the alternatives, we apply the most popular centroid-index ranking approach. We test the proposed approach by segmenting the suppliers of a medium-sized transport equipment company to illustrate its applicability. The company can thus formulate different strategies to handle various segments based on the outcomes obtained using the proposed method. We identify at least four major green supplier strategies: (i) maintain close long-term relationships with suppliers that have strong capabilities and high willingness [31]; (ii) improve and attract relationships with suppliers that have high capabilities, but a low-level willingness to cooperate [36]; (iii) help suppliers that have low capabilities, but are very willing "to green" their products and processes [35]; (iv) terminate relationships with suppliers that are less capable and less willing to cooperate [2, 3]. We further compare the proposed approach with another fuzzy MCDM approach to demonstrate its superiority. Findings show that the proposed approach is an effective tool for practitioners to solve GSS problems.

The study does have some limitations. First, the proposed approach does not consider the correlation of attributes. Therefore, it is difficult to derive the weights of the decision criteria while maintaining judgment consistency. Second, by using fuzzy sets, the proposed approach cannot handle MCGDM problems that have indeterminate and inconsistent information. Future work plans are to integrate an AHP method in MCGDM by defining the importance weights of criteria. Neutrosophic sets and their extension will also be applied to express the vague information in MCGDM.

## Supporting information

**S1 Appendix.**
(DOCX)

## Acknowledgments

This research was completed during and after the stay of Dr. Luu Quoc Dat at the Vietnam Institute for Advanced Study in Mathematics (VIASM).

## Author Contributions

**Conceptualization:** Do Anh Duc, Luu Huu Van, Vincent F. Yu, Shuo-Yan Chou, Ngo Van Hien, Dinh Van Toan, Luu Quoc Dat.

**Data curation:** Do Anh Duc, Luu Huu Van, Ngo Van Hien, Dinh Van Toan, Luu Quoc Dat.

**Formal analysis:** Do Anh Duc, Luu Huu Van, Ngo Van Hien, Dinh Van Toan, Luu Quoc Dat.

**Funding acquisition:** Luu Quoc Dat.

**Methodology:** Luu Huu Van, Shuo-Yan Chou, Luu Quoc Dat.

**Project administration:** Do Anh Duc.

**Resources:** Ngo Van Hien.

**Supervision:** Vincent F. Yu, Shuo-Yan Chou.

**Validation:** Vincent F. Yu, Shuo-Yan Chou, Luu Quoc Dat.

**Visualization:** Vincent F. Yu, Shuo-Yan Chou, Ngo Van Hien, Ngo The Chi, Dinh Van Toan, Luu Quoc Dat.

**Writing – original draft:** Do Anh Duc, Luu Huu Van, Dinh Van Toan, Luu Quoc Dat.

**Writing – review & editing:** Vincent F. Yu, Shuo-Yan Chou, Ngo The Chi, Luu Quoc Dat.

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
