## [Decision Letter · Decision Letter 0]

15 Jun 2020

PONE-D-20-03268

A DYNAMIC GENERALIZED FUZZY MULTI-CRITERIA GROUP DECISION MAKING APPROACH FOR GREEN SUPPLIER SEGMENTATION

PLOS ONE

Dear Dr. Luu Quoc,

Thank you for submitting your manuscript to PLOS ONE. After careful consideration, we feel that it has merit but does not fully meet PLOS ONE’s publication criteria as it currently stands. Therefore, we invite you to submit a revised version of the manuscript that addresses the points raised during the review process.

We look forward to receiving your revised manuscript.

Kind regards,

Yiming Tang, Ph.D.

Academic Editor

PLOS ONE

Journal Requirements:

Reviewers' comments:

Reviewer's Responses to Questions

**Comments to the Author**

1. Is the manuscript technically sound, and do the data support the conclusions?

Reviewer #1: Partly

Reviewer #2: Partly

2. Has the statistical analysis been performed appropriately and rigorously? 

Reviewer #1: Yes

Reviewer #2: I Don't Know

3. Have the authors made all data underlying the findings in their manuscript fully available?

Reviewer #1: Yes

Reviewer #2: Yes

4. Is the manuscript presented in an intelligible fashion and written in standard English?

Reviewer #1: Yes

Reviewer #2: No

5. Review Comments to the Author

Reviewer #1: I would like to thank the authors for this interesting approach in dealing with an important subject. The subject of segmentation and especially weighting using decision makers or "experts" are one of the areas of debate in many fields. I found this approach easy to follow and reproducible. This is an important advantage for the proposed method. However, I have few issues that I want to recommend and clarify:

1- I am not sure that you have presented enough discussion in the literature about the added value for your approach and the need for this approach, not only from operational point of view but also from computational one. In other words, you provided in the second paragraph of the introduction why you are proposing this approach, but there is no justification for where you think it will improve the current method statistically. Is this approach better in your opinion and why?!

2- Tables 6a - 6e are a bit confusing within the manuscript, maybe including these tables in an annex will be more convenient.

3- There must be a discussion section after the section "Comparison of the proposed method with another fuzzy

MCDM method" and the conclusion section. The results are not discussed clearly for the reader. The results need to be interpreted from mathematical and operational point of view, as a reader I am afraid I need more explanation for the numbers. It looked in some places that you jumped from section to another without explaining the results. Moreover, you need to discuss the strengths of your approach, how it tackled current existing problem, and why do you think it should be considered by others. For example, have you considered a simulation study and compare the results with other methods to assess the consistency of the results?! or have you considered comparing this method with more statistically based approaches such as Multidimensional Latent Class Item Response Theory Models?! There should be more discussion before you present your conclusion.

Reviewer #2: While new methods for "green" supplier segmentation is certainly important, interesting, and relevant, there are several issues in this paper.

The methods in this paper appear to be sound, it is very hard to read and comprehend. The organization and visualization of data/results is overall, poor. Additionally:

- Background on fuzzy numbers is lengthy and a bit hard to follow.

- There are an excess of tables, which is incredibly overwhelming and unhelpful given the complexity of the topic and notation. The tables in the literature review section are redundant or unnecessary. If tables really are necessary, for this many tables, they belong in an appendix.

- Some terms or abbreviations are not explained and confusing. For example in Table 6a, I'm assuming "fa" = "fair", "Ve_go" = "Very good"? This needs to be standardized and presented in a more meaningful, insightful, and visually interesting manner. For example, map responses to numbers rather than letter abbreviations, and plot a heat map of responses, rather than use a table. This can be done with ALL of the tables in this section

- Table 8 may be better off as some sort of visual representation (chart) rather than a table

- There are grammatical mistakes throughout the paper

6. PLOS authors have the option to publish the peer review history of their article (what does this mean?). If published, this will include your full peer review and any attached files.

Reviewer #1: No

Reviewer #2: No

---

## [Author Response · Author response to Decision Letter 0]

15 Sep 2020

Response to Referees’ Comments

PONE-D-20-03268

A DYNAMIC GENERALIZED FUZZY MULTI-CRITERIA GROUP DECISION MAKING APPROACH FOR GREEN SUPPLIER SEGMENTATION 

The authors greatly appreciate the time and effort the referees spent on reviewing this manuscript. This paper has been revised based on the constructive comments and suggestions made by the referees. Major changes are shown in red color.

Referee 1’s Comments:

I would like to thank the authors for this interesting approach in dealing with an important subject. The subject of segmentation and especially weighting using decision makers or "experts" are one of the areas of debate in many fields. I found this approach easy to follow and reproducible. This is an important advantage for the proposed method. However, I have few issues that I want to recommend and clarify:

1. I am not sure that you have presented enough discussion in the literature about the added value for your approach and the need for this approach, not only from operational point of view but also from computational one. In other words, you provided in the second paragraph of the introduction why you are proposing this approach, but there is no justification for where you think it will improve the current method statistically. Is this approach better in your opinion and why?

2. Tables 6a - 6e are a bit confusing within the manuscript, maybe including these tables in an annex will be more convenient.

3. There must be a discussion section after the section "Comparison of the proposed method with another fuzzy MCDM method" and the conclusion section. The results are not discussed clearly for the reader. The results need to be interpreted from mathematical and operational point of view, as a reader I am afraid I need more explanation for the numbers. It looked in some places that you jumped from section to another without explaining the results. Moreover, you need to discuss the strengths of your approach, how it tackled current existing problem, and why do you think it should be considered by others. For example, have you considered a simulation study and compare the results with other methods to assess the consistency of the results?! or have you considered comparing this method with more statistically based approaches such as Multidimensional Latent Class Item Response Theory Models?! There should be more discussion before you present your conclusion.

Responses:

1. Thank you very much for your comments. The authors have added some sentences in the introduction and literature review section to discuss more about the shortcomings of the existing approaches and the advantages of our approach. 

2. Thanks for your suggestion. The authors have moved Tables 6a - 6e to appendix section.

3. Thanks for your comments. The authors have added some paragraphs to discuss about the results of the study and the advantages of our approach. Some sentences have been added in the implementation section to explain more about the calculation process. In this study, a new dynamic generalized fuzzy MCDM approach has been proposed. Then, we have compared the proposed method with another fuzzy MCDM method to show its advantages. The comparison between our proposed approach with more statistically based approaches such as Multidimensional Latent Class Item Response Theory Models should be our further research. 

Referee 2’s Comments:

While new methods for "green" supplier segmentation is certainly important, interesting, and relevant, there are several issues in this paper. 

1. The methods in this paper appear to be sound, it is very hard to read and comprehend. The organization and visualization of data/results is overall, poor. 

2. Background on fuzzy numbers is lengthy and a bit hard to follow.

3. There is an excess of tables, which is incredibly overwhelming and unhelpful given the complexity of the topic and notation. The tables in the literature review section are redundant or unnecessary. If tables really are necessary, for this many tables, they belong in an appendix.

4. Some terms or abbreviations are not explained and confusing. For example, in Table 6a, I'm assuming "fa" = "fair", "Ve_go" = "Very good"? This needs to be standardized and presented in a more meaningful, insightful, and visually interesting manner. For example, map responses to numbers rather than letter abbreviations, and plot a heat map of responses, rather than use a table. This can be done with ALL of the tables in this section.

5. Table 8 may be better off as some sort of visual representation (chart) rather than a table.

6. There are grammatical mistakes throughout the paper

Responses:

1. Thank you very much for your comments. The authors have added some sentences in the implementation section to explain more about the data and results of this study. The authors have also moved the Tables 6a - 6e to the appendix.

2. Thanks for your suggestion. The authors have moved the background on fuzzy numbers to appendix.

3. Thanks for your suggestion. The authors have moved Tables 1-3 to the appendix.

4. Thanks for your suggestion. The authors have tried to change the abbreviations of linguistic variables (Appendix B - Table 2 and other tables). 

5. Thanks for your suggestion. The authors have modified the Table 8 to make it more visually.

6. The authors have tried to fix the grammatical mistakes throughout the paper.

The authors would like to thank again the reviewers for the time and expertise they have invested in these reviews. The revised manuscript with marked changes has been resubmitted to your journal. We look forward to your positive response.

Sincerely,

Luu Quoc Dat

---

## [Decision Letter · Decision Letter 1]

26 Dec 2020

A DYNAMIC GENERALIZED FUZZY MULTI-CRITERIA GROUP DECISION MAKING APPROACH FOR GREEN SUPPLIER SEGMENTATION

PONE-D-20-03268R1

Dear Dr. Luu Quoc,

We’re pleased to inform you that your manuscript has been judged scientifically suitable for publication and will be formally accepted for publication once it meets all outstanding technical requirements.

Kind regards,

Yiming Tang, Ph.D.

Academic Editor

PLOS ONE

Additional Editor Comments (optional):

Reviewers' comments:

Reviewer's Responses to Questions

**Comments to the Author**

1. If the authors have adequately addressed your comments raised in a previous round of review and you feel that this manuscript is now acceptable for publication, you may indicate that here to bypass the “Comments to the Author” section, enter your conflict of interest statement in the “Confidential to Editor” section, and submit your "Accept" recommendation.

Reviewer #1: All comments have been addressed

Reviewer #3: All comments have been addressed

Reviewer #4: All comments have been addressed

2. Is the manuscript technically sound, and do the data support the conclusions?

Reviewer #1: Yes

Reviewer #3: Partly

Reviewer #4: Yes

3. Has the statistical analysis been performed appropriately and rigorously? 

Reviewer #1: Yes

Reviewer #3: Yes

Reviewer #4: Yes

4. Have the authors made all data underlying the findings in their manuscript fully available?

Reviewer #1: Yes

Reviewer #3: Yes

Reviewer #4: Yes

5. Is the manuscript presented in an intelligible fashion and written in standard English?

Reviewer #1: Yes

Reviewer #3: Yes

Reviewer #4: Yes

6. Review Comments to the Author

Reviewer #1: I believe all the comments have been addressed by the authors. I reviewed the new version and I find it now easier to follow and more visually appealing. I encourage the authors to do further investigation in the subject and do comparison with more statistically established methods in their future work.

Reviewer #3: The Author should support with evidence during discussion of results. The Author has good presentation but poor discussion on the results found.

Reviewer #4: (No Response)

7. PLOS authors have the option to publish the peer review history of their article (what does this mean?). If published, this will include your full peer review and any attached files.

Reviewer #1: No

Reviewer #3: **Yes: **Zerihun Yohannes Amare

Reviewer #4: **Yes: **Dr. Ghaffar Ali

---

## [Editor Report · Acceptance letter]

13 Jan 2021

PONE-D-20-03268R1 

A Dynamic Generalized Fuzzy Multi-Criteria Croup Decision Making Approach for Green Supplier Segmentation 

Dear Dr. Dat:

I'm pleased to inform you that your manuscript has been deemed suitable for publication in PLOS ONE. Congratulations! Your manuscript is now with our production department. 

Kind regards, 

on behalf of

Professor Yiming Tang 

Academic Editor

PLOS ONE